# Effects of Different Capsulotomy and Fragmentation Energy Levels on the Generation of Oxidative Stress Following Femtosecond Laser-Assisted Cataract Surgery

**DOI:** 10.3390/biom14030318

**Published:** 2024-03-07

**Authors:** Sang Beom Han, Yu-Chi Liu, Melina Setiawan, Isabelle Xin Yu Lee, Moushmi Patil, Hon Shing Ong, Jodhbir S. Mehta

**Affiliations:** 1Saevit Eye Hospital, Goyang 10447, Republic of Korea; sbhan@kangwon.ac.kr; 2Tissue Engineering and Cell Therapy Group, Singapore Eye Research Institute, Singapore 169856, Singapore; melina.setiawan@seri.com.sg (M.S.); hirmess@daum.net (I.X.Y.L.); ong.hon.shing@singhealth.com.sg (H.S.O.); jodhbir.s.mehta@singhealth.com.sg (J.S.M.); 3Singapore National Eye Centre, Singapore 168751, Singapore; 4Ophthalmology Academic Clinical Program, Duke-NUS Graduate Medical School, Singapore 169857, Singapore

**Keywords:** cataract surgery, femtosecond laser, femtosecond laser-assisted cataract surgery, oxidative stress

## Abstract

Purpose. This study was conducted to evaluate the effects of different capsulotomy and fragmentation energy levels on the production of oxidative free radicals following femtosecond laser-assisted cataract surgery (FLACS) with a low-energy platform. Methods. The experimental study included 60 porcine eyes (12 groups). In each group, capsulotomies with 90% or 150% energy, and fragmentations with 90%, 100%, or 150% energy or 150% with high spot density, respectively, were performed. Control samples were obtained from non-lasered eyes at the beginning (five eyes) and end (five eyes) of the experiment. In the clinical study, 104 eyes were divided into 5 groups, and they received conventional phacoemulsification (20 eyes), FLACS with 90% capsulotomy and 100% fragmentation energy levels without NSAIDs (16 eyes), FLACS with 90% (26 eyes) or 150% (22 eyes) capsulotomy energy levels, respectively, with a 100% fragmentation energy level and NSAIDs, and FLACS with 90% capsulotomy and 150% fragmentation energy levels and NSAIDs (20 eyes). Aqueous samples were analyzed for their malondialdehyde (MDA) and superoxide dismutase (SOD) levels. Results. In the experimental study, there were no significant differences in the MDA and SOD levels between the groups with different capsulotomy energy levels. An increase in the fragmentation energy from 100% to 150% led to significantly higher MDA levels in the groups with both 90% (*p* = 0.04) and 150% capsulotomy energy levels (*p* = 0.03), respectively. However, increased laser spot densities did not result in significant changes in MDA or SOD levels. In the clinical study, all four of the FLACS groups showed higher MDA levels than the conventional group. Similarly, the increase in the fragmentation energy from 100% to 150% resulted in significantly elevated levels of MDA and SOD, respectively. Conclusions. Although increasing the FSL capsulotomy energy level may not have increased free radicals, higher fragmentation energy levels increased the generation of aqueous free radicals. However, fragmentation with high spot density did not generate additional oxidative stress. Increased spot density did not generate additional oxidative stress, and this can be helpful for dense cataracts.

## 1. Introduction

Femtosecond laser-–assisted cataract surgery (FLACS) has been shown to produce more predictable and precise capsulotomies, per Refs. [1,2,3], although the visual and refractive outcomes are suggested to be equivocal compared to conventional cataract surgery [4,5,6,7]. Among the five commercially available femtosecond laser (FSL) platforms, the Femto LDV Z8 (Ziemer Ophthalmic Systems AG, Port, Switzerland) is the only high-frequency and low-energy system [1,8,9]. This can be advantageous due to its increased laser precision with respect to laser cutting and the reduction in collateral tissue damage [10,11,12]. In our previous randomized controlled study, FLACS using a low-energy platform showed significantly reduced endothelial cell count (ECC) losses compared to conventional cataract surgery, even though their visual acuity results were comparable [13].

Phacoemulsification is associated with the increased generation of oxygen free radicals during nuclear disassembly, and this can result in ECC reductions caused by oxidative damage [13]. However, the association between FLACS and aqueous oxidative stress has not been extensively investigated. In 2019, Masuda et al. [14] showed that lens irradiation with a high-energy FSL platform (CATALYS, Johnson & Johnson Surgical Vision, Inc. Santa Ana, CA, USA) increased the production of aqueous free radicals in proportion to the amount of laser energy, indicating that FLACS can be associated with the increased generation of intracameral free radicals. The authors also suggested that excessive laser irradiation during FLACS might cause increased corneal endothelial damage caused by the free radicals [14]. However, this was an ex vivo study in porcine eyes, and it did not take into consideration free radical formation created by the phacoemulsification itself.

We have previously shown that laser pre-treatment with a low-energy FLACS platform led to non-statistically significant increases in the levels of oxidative free radicals compared to a conventional surgery [8]. However, phacoemulsification itself resulted in an increase in free radicals that was proportional to the effective phacoemulsification time (EPT) [8]. In addition, there were no significant differences in the levels of aqueous free radicals between the FLACS and conventional groups at the end of surgery, suggesting that the increased oxidative stress caused by the FSL pre-treatment was negated following phacoemulsification [8].

For a better understanding of the influence of FLACS on aqueous oxidative stress, the changes in the levels of aqueous oxidative free radicals at each step of the laser pre-treatment, i.e., capsulotomy and lens fragmentation, as well as the changes in the levels of the free radicals in relation to the different levels of laser energy, should be studied. Therefore, in the present study, we investigated the effects of different FSL energy levels during capsulotomies and fragmentations, respectively, on the generation of oxidative free radicals in FLACS with a low-energy platform. We also compared these results with those from conventional surgeries.

## 2. Patients and Methods

This study was approved by the Institutional Review Board of SingHealth (IRB No. 2015/2565), Singapore, and it was conducted in accordance with the Declaration of Helsinki.

### 2.1. Experimental Study

An experimental study using porcine eyes was conducted for the evaluation of the effect of different levels of lens fragmentation energy, in addition to FSL capsulotomy, on the generation of oxidative free radicals. A total of 60 freshly enucleated porcine eyes (post-mortem time < 8 h), obtained from a local abattoir (Primary Industries Pte Ltd., Singapore) and submerged in Optisol (Bausch & Lomb, Berlin, Germany) to prevent corneal swelling, per Ref. [15], were randomly allocated to 10 sub-groups (*n* = 5 for each group). In two groups (groups A and B), anterior capsulotomies with 90% or 150% energy without fragmentation were performed. In six groups, anterior capsulotomies with 90% or 150% energy, respectively, and lens fragmentation with 90%, 100% or 150% energy, respectively, were performed. In the remaining two groups, increased laser spot density was applied. Aqueous humor samples (100 to 150 μL) were obtained using a 30-gauge needle attached to a 1 cc syringe through a limbal paracentesis after the laser pre-treatment. The porcine aqueous humor samples were immediately transferred onto dry ice and transported to a laboratory where they were stored at −80 °C until analysis. To reduce inter-experiment variation, control aqueous samples were also taken from non-lasered eyes at the beginning (5 eyes) and end (5 eyes) of the experiment sessions (Table 1).

### 2.2. Clinical Study

In this clinical study, a total of 104 eyes (104 patients; mean age, 67.8 ± 8.2) were included. In 20 eyes, conventional phacoemulsification was performed without pre-treatment with topical non-steroidal anti-inflammatory drugs (NSAIDs) (0.4% topical ketorolac tromethamine [Allergan, Inc., Irvine, CA, USA] four times daily for 1 day before surgery (group A). Sixteen and twenty-six eyes were assigned to FLACS groups with 90% capsulotomy and 100% fragmentation energy levels, respectively, and with or without pre-treatment with topical NSAIDs, respectively (groups B and C). In group D, 22 eyes received FLACS with 150% capsulotomy and 100% fragmentation energy levels with pre-treatment and with NSAIDs. In group E, 20 eyes underwent FLACS with 90% capsulotomy and 150% fragmentation energy levels with NSAID pre-treatment.

The exclusion criteria were as follows: (1) pupil size of <6 mm, (2) pseudoexfoliation syndrome, (3) floppy iris syndrome, (4) history of the use of systemic or topical NSAIDs within 1 month, (5) allergy to NSAIDs, (6) rheumatic diseases, and (7) ocular inflammatory disease.

For mydriasis, 0.5% tropicamide (Alcon Laboratories, Inc., Fort Worth, TX, USA) and 2.5% phenylephrine hydrochloride (Bausch & Lomb, Berlin, Germany) eye drops were instilled three times within 1 h preoperatively. All cataract surgeries were performed under local anesthesia with topical proparacaine hydrochloride (Alcaine^®^; Norvatis, Basel, Switzerland) and a peribulbar block with sedation.

For group A, a 2.7 mm corneal incision was created with a keratome, and conventional phacoemulsification surgery was performed with the Infiniti Vision Ozil system (Alcon Laboratories, Inc., Fort Worth, TX, USA). For the FLACS groups (groups B, C, D, and E), corneal incisions, anterior capsulotomy, and lens fragmentation was performed with an LDV Z8 femtosecond laser system (Ziemer Ophthalmic Systems AG). After the patient interface was docked, the suction was applied to the eye, followed by docking of the laser machine handpiece. The procedures were performed with the same parameters as in our previous studies (capsulotomy diameter of 5 mm, cutting depth of 0.4 mm, repetition rate of 1 MHz, pulse duration of 250 fs, capsulotomy energy level of 90% (groups B, C, and E) or 150% (group D), angulation between the handpiece and Z8 moveable arm of −10°, and a suction pressure of 400 mbar) (refs. [11,16]). Lens fragmentation with a six-piece pie-cut pattern and a fragmentation energy of 100% (groups B, C, and D) or 150% (group E) [16] was performed, followed by a 2.6 mm corneal incision. In the 4 FLACS groups, phacoemulsification and intraocular lens (Acrysof SA60AT; Alcon Laboratories, Inc.) implantation was performed in the same manner as in group A. In all patients, sodium hyaluronate 3.0% with chondroitin sulfate 4.0% (Viscoat; Alcon Laboratories, Inc.) was used as an OVD [11]. After the cataract surgery, topical 0.1% preservative-free dexamethasone and 0.5% levofloxacin eye drops were prescribed six times a day for one week, and this was tapered to four times a day for four weeks.

For all eyes, aqueous humor samples (100 to 150 μL) were collected using a 30-gauge needle attached to a 1 cc syringe through limbal paracentesis before and after phacoemulsification, respectively. In group A, aqueous humor samples were obtained before the creation of a corneal main wound and at the end of the phacoemulsification. In groups B, C, D, and E, the aqueous samples were collected within 5 min after the laser pre-treatment and at the end of the phacoemulsification. Each aqueous sample was immediately transferred onto dry ice and transported to a laboratory, where they were stored at −80 °C until analysis.

### 2.3. Quantification of Aqueous Free Radicals

An aqueous malondialdehyde (MDA) enzyme-linked immunosorbent assay (ELISA) kit (MyBiosource, Inc. San Diego, CA, USA) and a superoxide dismutase (SOD) enzyme-linked immunosorbent assay (ELISA) kit (MyBiosource, Inc.) were used according to the manufacturers’ instructions. Aqueous MDA and SOD levels were analyzed in both the clinical and experimental studies, respectively.

## 3. Statistical Analysis

Statistical analysis was performed using SPSS software (ver. 21.0; SPSS, Inc., Chicago, IL, USA). For the clinical study, the sample size required was calculated using the preliminary MDA results of the first three patients, as follows: 350.2 ± 158.3 and 518.2 ± 158.3 ng/mg protein for group A and group B, respectively. A sample size of 14 would be required to confirm the difference with a power of 80% and a significance level of 5% in the clinical study. All continuous variables were expressed as means ± standard deviations. In both the clinical and porcine studies, a one-way analysis of variance (ANOVA) with a Bonferroni correction was applied for the comparisons between the groups. A paired *t*-test was used to compare the MDA values before and after the phacoemulsification in the clinical study. A *p* value of <0.05 was considered statistically significant.

## 4. Results

### 4.1. Experimental Study

Between the groups receiving capsulotomy energy levels of 90% and 150% with no fragmentation, there were no significant differences in the MDA (*p* = 0.29) and SOD (*p* = 0.62) levels. In comparison with the groups receiving various fragmentation energy levels with a capsulotomy energy level of 90%, the MDA and SOD concentrations increased with the increases in the fragmentation energy levels. The group with a fragmentation energy level of 150% showed a significantly higher MDA level than the group with a fragmentation energy level of 100% (*p* = 0.04) (Figure 1A). Similarly, in comparison with the groups receiving various fragmentation energy levels with a capsulotomy energy level of 150%, the MDA and SOD levels increased when there was an increase in the fragmentation energy level. The group with a fragmentation energy level of 150% showed significantly higher MDA levels than the group with a fragmentation energy level of 100% (*p* = 0.03). Similar trends and changes were also seen in the SOD changes, although they did not reach statistical significance (Figure 1B). Increasing the laser spot density did not result in significant changes in the MDA or SOD levels (Figure 1).

### 4.2. Clinical Study

After capsulotomies with FSL, there were significant differences in the MDA levels among the five groups (*p* = 0.047, ANOVA). All the FSL groups (groups B, C, D, and E) showed significantly higher aqueous MDA levels compared to the conventional group (group A) (372.0 ± 171.5, 472.4 ± 213.8, 531.8 ± 253.6, 458.3 ± 179.7, and 644.1 ± 199.5 (ng/mg protein) for groups A, B, C, D, and E, respectively; *p* = 0.02, *p* < 0.01, *p* = 0.01, and *p* = 0.006 for groups A vs. B, A vs. C, A vs. D, and A vs. E, respectively). There were no significant differences in the MDA levels between any two of the FSL groups with 100% fragmentation energy levels (groups B, C, and D), respectively (*p* = 0.11, *p* = 0.28, and *p* = 0.09 for groups B vs. C, B vs. D, and C vs. D, respectively). However, the group with a 150% fragmentation energy level (group E) showed significantly higher MDA levels than groups B and D (*p* = 0.02 and *p* = 0.03, respectively). Group E also showed higher MDA levels than group C, although the differences did not reach statistical significance (*p* = 0.05) (Figure 2A).

Regarding the SOD levels, the FSL groups with 100% fragmentation energy levels did not show significantly different SOD levels compared to the conventional group (7.64 ± 1.88, 8.38 ± 2.98, 8.68 ± 3.12, and 8.42 ± 5.58 (ng/mg protein) for groups A, B, C, and D, respectively; *p* =0.31, *p* = 0.28, and *p* = 0.30 for groups A vs. B, A vs. C, and A vs. D, respectively). However, the group with a 150% fragmentation energy level (group E; 10.70 ± 3.05 ng/mg protein) showed significantly higher SOD levels than the conventional group (group A) and the groups with 100% fragmentation energy levels (groups B and D) (*p* = 0.007, *p* = 0.03, and *p* = 0.04, respectively).

Of note, there were no significant differences in the MDA and SOD levels between groups B and C (*p* = 0.11 and *p* = 0.67, respectively), suggesting that the preoperative use of NSAIDs did not prevent the generation of oxidative stress (Figure 2B).

## 5. Discussion

We showed that an increase in FSL fragmentation energy—not capsulotomy—was associated with an elevation in oxygen free radical levels in a porcine experimental study. The clinical study confirmed the results and also showed that an increase in fragmentation was associated with increases in intracameral MDA and SOD levels compared to conventional surgery. However, increasing the spot density of laser fragmentation was not associated with significant changes in MDA or SOD levels. Elevated FSL capsulotomy energy levels did not lead to elevations in aqueous free radical levels in both the porcine and human studies. The pre-operative administration of NSAID treatment did not suppress the MDA and SOD increases resulting from the laser capsulotomy.

We used aqueous concentrations of MDA and SOD for the evaluation of intracameral free radical activity because these biomarkers have been used for the measurement of intraocular oxidative stress, which is associated with various ocular diseases [17,18,19,20]. Previous studies have shown that free radicals can be generated by ultrasound power in the presence of aerobic solutions in proportion to the effective phacoemulsification time that reflects the amount and duration of the ultrasound power [21,22]. Phacoemulsification in an aqueous solution generates cavitation bubbles that lead to the dissociation of water molecules which facilitate the production of free radicals [23]. Takahashi et al. [21] showed that free radicals can also be produced even with irrigation and aspiration [21]. These free radicals may cause disruption of the blood–aqueous barrier, which results in the increased production of inflammatory cytokines and prostaglandins and increased endothelial cell loss [23,24].

Using a high-energy FSL system (CATALYS; Johnson & Johnson Surgical Vision Inc., Santa Ana, CA, USA), Masuda et al. [14] showed that an increased production of free radicals was seen with lens irradiation with higher laser energies. In their study, an increase in FSL irradiation total energy from 20 J to 40 J resulted in a 1.5-fold increase in the intracameral free radical level [14]. In the current study, we showed that FSL capsulotomy using a low-energy FSL system did not result in the increased generation of intracameral free radicals, and an increase in the capsulotomy energy level did not lead to further increases in free radicals. An increase in the capsulotomy energy level is useful in the case of capsulotomy formation in the presence of an OVD in the anterior chamber [25,26]. Increases in the FSL capsulotomy energy levels did lead to increased concentrations of pro-inflammatory IL-8 [11]. However, there was no increased generation of aqueous free radicals [11]. These findings suggest that increasing capsulotomy energy levels does not lead to additional oxidative stress in the anterior chamber.

The results from both the porcine ex vivo study and the human clinical study showed that increasing the laser fragmentation energy level from 100% to 150% resulted in the increased generation of aqueous free radicals, both in the MDA (approximately 30%) and SOD (approximately 25%) levels, irrespective of the capsulotomy energy level. These results suggest that laser pre-treatment with a low-energy FSL platform results in increased oxidative stress in the aqueous humor, and a 1.5-fold increase in the laser fragmentation energy level may lead to a 25–30% increase in the generation of aqueous free radicals. These findings were supported by our clinical study that showed higher pre-phacoemulsification MDA levels in all of the femtosecond laser groups compared to conventional surgery. Although the Femto LDV Z8 low-energy system generates fewer cavitation bubbles during fragmentation due to its substantially reduced pulsed energy level (in the <5 nJ range) compared to the high-energy system (8–10 μJ for the CATALYS system), per Ref. [27], increasing the laser fragmentation power can still increase the amount of cavitation bubbles that leads to the increased generation of aqueous free radicals [8,28].

We previously showed that there were no significant differences in the aqueous free radical levels between the FLACS and conventional groups after phacoemulsification, suggesting that the increased generation of oxidative stress caused by the FSL fragmentation could be counterbalanced with a decreased need for phacoemulsification energy [8]. An increase in free radicals during phacoemulsification has been shown to correlate with EPT, per Ref. [8], and increasing the fragmentation energy during FLACS in dense or mature cataracts has been shown to reduce EPT [29,30]. Hence, the result was the decreased generation of free radicals during phacoemulsification, which could compensate for the increased free radicals during the FSL fragmentation. This may explain the better ECC in patients post-FLACS in cases with lower EPT [16]. We further tested high-density fragmentation that appeared, in our experience, to be helpful in cases with the highest-grade cataracts. The ex vivo results suggested there were no further increases in the MDA and SOD levels, and hence, this higher density setting could be safely applied in such cases without unwanted increases in free radical formation. This was likely because the total energy generated from the low-energy FSL system was <1–5 J, which was substantially lower compared to that used in the study by Masuda et al. [14]. Moreover, with low energy, high pulses, and overlapping spots, there can be no further free radical formation beyond a certain threshold since free radicals cannot be generated in gas and can only occur when interacting with tissue.

The present study has limitations. The association between lens density and the amount of generated free radicals during laser fragmentation could not be completely ascertained since only young porcine eyes were used in the experimental study for the changes in the MDA and SOD levels according to the different fragmentation energies, and only 100% and 150% fragmentation energy levels were evaluated in the clinical study. In addition, with modern phacoemulsification techniques, lens density has a lesser correlation with EPT.

In summary, this study showed that an increase in FSL capsulotomy energy was not associated with increases in aqueous free radical levels. Increasing the laser fragmentation energy levels caused increases in aqueous free radicals, although this could be counterbalanced by reducing the phacoemulsification energy caused by the FSL fragmentation. Fragmentation with high FSL spot density did not generate additional oxidative stress, and it can be safely applied in dense cataracts without further increases in free radical formation.

## Figures and Tables

**Figure 1 biomolecules-14-00318-f001:**
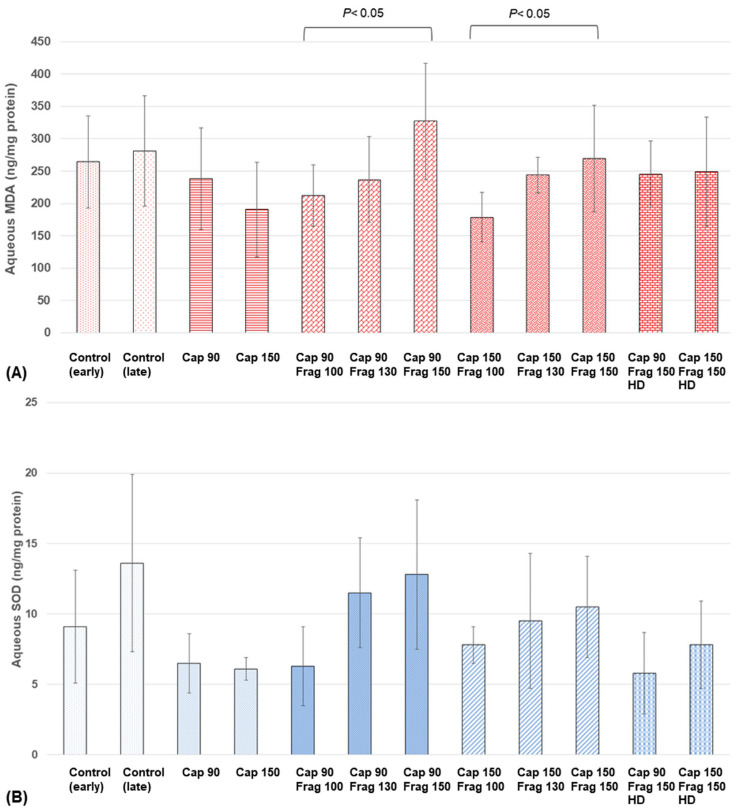
Aqueous oxidative stress levels at different capsulotomy and lens fragmentation energy levels in the experimental study (Cap, capsulotomy; Frag, fragmentation). (**A**) Aqueous MDA after femtosecond laser (FSL) pre-treatment with various capsulotomy and fragmentation energy levels. (**B**) Aqueous SOD after FSL pre-treatment with various energy levels.

**Figure 2 biomolecules-14-00318-f002:**
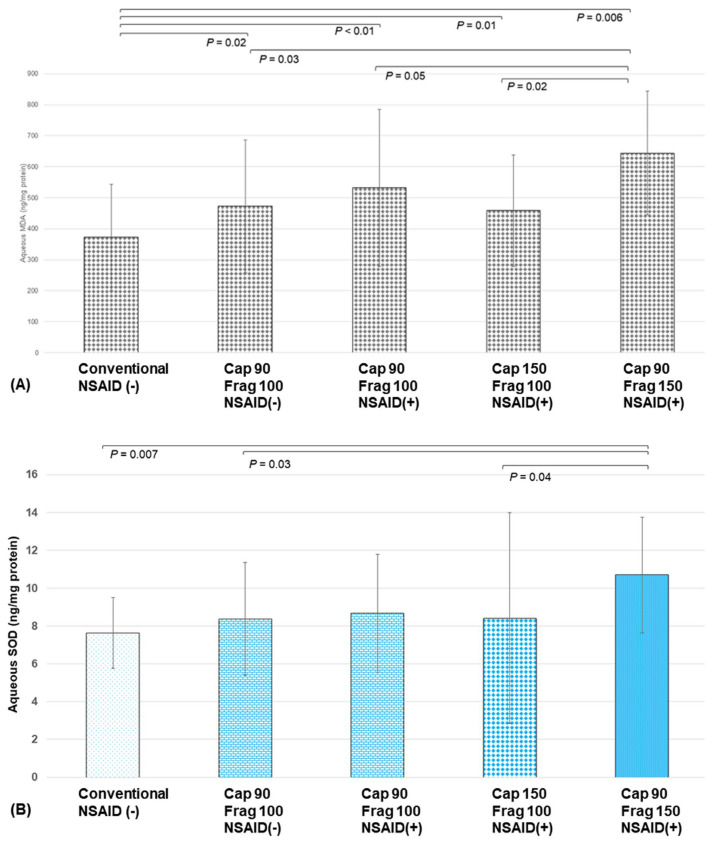
Aqueous oxidative stress levels at the different capsulotomy and fragmentation energy levels in the clinical study (Cap, capsulotomy; Frag, fragmentation). (**A**) Aqueous MDA after femtosecond laser (FSL) pre-treatment with various capsulotomy and fragmentation energy levels. (**B**) Aqueous SOD after FSL pre-treatment with various energy levels.

**Table 1 biomolecules-14-00318-t001:** Groups of the experimental eyes studied with various capsulotomy and fragmentation energy levels, respectively (n = 5 porcine eyes for each group).

Group	Capsulotomy Energy	Fragmentation Energy
Control (early)	None	None
Control (late)	None	None
Cap 90	Capsulotomy energy 90%	None
Cap 150	Capsulotomy energy 150%	None
Cap 90, frag 100	Capsulotomy energy 90%	Fragmentation energy 100% Standard laser spot density
Cap 90, frag 130	Capsulotomy energy 90%	Fragmentation energy 130% Standard laser spot density
Cap 90, frag 150	Capsulotomy energy 90%	Fragmentation energy 150% Standard laser spot density
Cap 150, frag 100	Capsulotomy energy 150%	Fragmentation energy 100% Standard laser spot density
Cap 150, frag 130	Capsulotomy energy 150%	Fragmentation energy 130% Standard laser spot density
Cap 150, frag 150	Capsulotomy energy 150%	Fragmentation energy 150% Standard laser spot density
Cap 90, frag 150, high spot density (HD)	Capsulotomy energy 90%	Fragmentation energy 150% Higher laser spot density
Cap 150, frag 150, HD	Capsulotomy energy 150%	Fragmentation energy 150% Higher laser spot density

## Data Availability

The data presented in this study are available on request from the corresponding author. The data are not publicly available due to privacy of the study subjects.

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
