# Peer review of "Effects of Different Capsulotomy and Fragmentation Energy Levels on the Generation of Oxidative Stress Following Femtosecond Laser-Assisted Cataract Surgery"

_biomolecules, 2024, doi:10.3390/biom14030318_

Round 1

Reviewer 1 Report

Comments and Suggestions for Authors

The presented article offers a sound and valuable approach to the problem of generation of oxidative stress during high-energy femto-second laser-assisted cataract surgery. The experimental design, materials and methods used and the discussion were appropriate and concise. What lacks in the presented data in the clinical study is the age of the patients.

The presentation of the results might be improved as well. Figures look unattractive and captions - hard to read; it could be easily corrected with little to no efforts, but will improve the overall quality of the manuscript.

The evaluation of the effect of NSAIDS has a confirmatory character, in accordance with the results already presented by Hecht et al., 2020.

Author Response

The presented article offers a sound and valuable approach to the problem of generation of oxidative stress during high-energy femto-second laser-assisted cataract surgery. The experimental design, materials and methods used and the discussion were appropriate and concise. What lacks in the presented data in the clinical study is the age of the patients.

 ⇒ The mean age is 67.8 ± 8.2. We mentioned it in the revised manuscript (Line 96).

 The presentation of the results might be improved as well. Figures look unattractive and captions - hard to read; it could be easily corrected with little to no efforts, but will improve the overall quality of the manuscript. The evaluation of the effect of NSAIDS has a confirmatory character, in accordance with the results already presented by Hecht et al., 2020.

 ⇒ As the reviewer suggested, we have improved the figures 1 and 2 (including the size and resolution of the captions). We also believe it can improve the overall quality of the manuscript.

⇒ Thank you very much for your encouraging and supportive comments. 

Reviewer 2 Report

Comments and Suggestions for Authors

The authors present an interesting paper on the generation of oxidative stress following fs-laser surgery. The paper is well written and presents original data. There are two major parts in this study: an experimental study with porcine eyes, (split into 10 subgroups) and an clinical study with n=104 eyes (spit into 5 subgroups). A number of clarifications would be wellcome:

- Patients and methods: is there a specific reason why the authors split the porcine eyes into10 different subgroups, whereas the clinical group was split into 5 subgroups? Why not using the same parameters in both arms of the study?

- 3. Statistical Analysis: The authors only describe the statistic analysis ot the clinical study - the statement "A sample size of 14 would be required" contradicts the sample size of n=5 in the experimental study.

-Results: in the discussion the authors state correctly that the increased energy in laser fragmentation can be counterbalanced by reduced phacoemulsification energy. As phacoemulsification was actually performed in these eyes, it should be possible to give the data of the needed phaco energy to prove this point.

Author Response

 - Patients and methods: is there a specific reason why the authors split the porcine eyes into10 different subgroups, whereas the clinical group was split into 5 subgroups? Why not using the same parameters in both arms of the study?

 ⇒ We thank you for the comments. Several subgroups, such as “control groups without further procedure”, or “capsulotomy only without fragmentation” were feasible in the experimental settings with the porcine eyes but not feasible in the clinical study because of ethical and practical reasons. For example, it is not quite necessary to have ‘early and late control groups’ in which no procedure was done and only aqueous sample was collected. Moreover, it was not possible to do capsulotomy only with without further fragmentation in the real-world cataract surgery.  Additionally, from the porcine study, we understood the MDA changes mainly came from fragmentation energy. Hence, we focused on the groups with different fragmentation energy and those were clinically relevant in the clinical study, even though these subgroups of laser settings were available.   

3. Statistical Analysis: The authors only describe the statistic analysis ot the clinical study - the statement "A sample size of 14 would be required" contradicts the sample size of n=5 in the experimental study.

⇒ We thank you for the comment. The porcine study was performed as an experimental study, and the sample size we described in the manuscript was calculated for the clinical study. The required sample number was calculated based on the preliminary MDA results of the first three patients: 350.2 ± 158.3 and 518.2 ± 158.3 ng/mg protein for the groups A and B in the clinical study, respectively. Thus, the statement "A sample size of 14 would be required" does not contradict the sample size of n=5 in the experimental study. To make this clearer, in the revised manuscript, we clearly described that the sample size calculated was for clinical study to avoid misunderstanding (Lines 146-151).  

 -Results: in the discussion the authors state correctly that the increased energy in laser fragmentation can be counterbalanced by reduced phacoemulsification energy. As phacoemulsification was actually performed in these eyes, it should be possible to give the data of the needed phaco energy to prove this point.

⇒ Thank you for your comment. We did not record phacoemulsification energy in this study. However, we have previously shown that the induction of MDA was associated with phacoemulsification time, which indirectly supports this point. We have revised the manuscript as follows: “We have previously shown that there was no significant difference in aqueous free radical levels between FLACS and conventional groups after phacoemulsification, suggesting that the increased generation of oxidative stress caused by FSL fragmentation could be counterbalanced with decreased need for phacoemulsification energy” (Lines 256-257).

⇒Thank you very much for the supportive and pertinent comments.